# Cost-Effectiveness of Influenza Vaccination Strategies in Adults: Older Adults Aged ≥65 Years, Adults Aged 50–64 Years, and At-Risk Adults Aged 19–64 Years

**DOI:** 10.3390/vaccines10030445

**Published:** 2022-03-14

**Authors:** Min Joo Choi, Gyeongseon Shin, Daewon Kang, Jae-Ok Lim, Yun-Kyung Kim, Won Suk Choi, Jae-Won Yun, Ji Yun Noh, Joon Young Song, Woo Joo Kim, Sang-Eun Choi, Hee Jin Cheong

**Affiliations:** 1Department of Internal Medicine, International St. Mary’s Hospital, Catholic Kwandong University, Incheon 22711, Korea; cowgow@naver.com; 2College of Pharmacy, Korea University, Sejong 30019, Korea; gyeongseon4265@gmail.com (G.S.); dwkang85@gmail.com (D.K.); 3Department of Data-Centric Problem Solving Research, Korea Institute of Science and Technology Information, Daejeon 34141, Korea; bounne@korea.ac.kr; 4Department of Pediatrics, Korea University Ansan Hospital, Korea University College of Medicine, Ansan 15355, Korea; byelhana@korea.ac.kr; 5Department of Internal Medicine, Korea University Ansan Hospital, Korea University College of Medicine, Ansan 15355, Korea; cmcws@daum.net; 6Department of Internal Medicine, Korea University Guro Hospital, Korea University College of Medicine, Seoul 08308, Korea; dr.jwyun@gmail.com (J.-W.Y.); jynoh@korea.ac.kr (J.Y.N.); infection@korea.ac.kr (J.Y.S.); wjkim@korea.ac.kr (W.J.K.)

**Keywords:** influenza, vaccine, cost-effectiveness, strategy, adult, older adults

## Abstract

The high disease burden of influenza in elderly and chronically ill adults may be due to the suboptimal effectiveness and mismatch of the conventional trivalent influenza vaccine (TIV). This study evaluated the cost-effectiveness of quadrivalent (QIV), adjuvanted trivalent (ATIV), and high-dose quadrivalent (HD-QIV) vaccines versus TIV used under the current Korean National Immunization Program (NIP) in older adults aged ≥65 years. We also evaluated the cost-effectiveness of programs for at-risk adults aged 19–64 and adults aged 50–64. A one-year static population model was used to compare the costs and outcomes of alternative vaccination programs in each targeted group. Influenza-related parameters were derived from the National Health Insurance System claims database; other inputs were extracted from the published literature. Incremental cost-effectiveness ratios (ICERs) were assessed from a societal perspective. In the base case analysis (older adults aged ≥65 years), HD-QIV was superior, with the lowest cost and highest utility. Compared with TIV, ATIV was cost-effective (ICER $34,314/quality-adjusted life-year [QALY]), and QIV was not cost-effective (ICER $46,486/QALY). The cost-effectiveness of HD-QIV was robust for all parameters except for vaccine cost. The introduction of the influenza NIP was cost-effective or even cost-saving for the remaining targeted gr3oups, regardless of TIV or QIV.

## 1. Introduction

Although many consider influenza a mild, self-limiting viral illness, it represents a serious public health problem because of the accompanying pneumonia and high mortality among the at-risk population [1]. Furthermore, it can cause a considerable socioeconomic impact through reduced workplace productivity and absenteeism during the infection epidemic [2].

Annual influenza vaccination is the most effective strategy to reduce the burden of influenza. For this reason, virtually all industrialized and many developing countries recommend annual vaccinations for high-risk populations [3,4,5,6]. The main strategies differ slightly from country to country, but most nations recommend influenza vaccination for older adults, young children, pregnant women, and chronically ill patients with high priority. In addition, some countries provide vaccines free of charge to those aged ≥ 50 years by expanding the age range of older adults [3,4,5,6,7], while others preferentially encourage vaccination of school-age children because of their role in disease transmission [4,5,7]. 

The Korean National Immunization Program (NIP) also focuses on the groups for whom influenza vaccination is particularly recommended. They include the elderly, children aged between six months and 18 years, and pregnant women [8]. Currently, influenza NIP in South Korea is being expanded step-by-step; however, some problems remain unsolved. First, the disease burden remains high among older adults despite a high vaccination rate (~85%) [8,9]. A major consideration is the problem of low immunogenicity among older adults [10]. To resolve this issue, the landscape of available influenza vaccines is continuously changing, and highly immunogenic influenza vaccines, such as high-dose (HD) and MF59-adjuvanted vaccines, have been developed and licensed [3,4]. Although data on the efficacy of these vaccines are limited, they are expected to be superior to preexisting standard-dose non-adjuvanted vaccines [11,12,13,14,15,16]. Consequently, to improve vaccination strategies for older adults in South Korea, it is necessary to preemptively evaluate the cost-effectiveness of these superior immunogenic influenza vaccines compared with conventional vaccines. Although such highly immunogenic influenza vaccines are not commercially available in South Korea, they are expected to be introduced soon. In addition, the Korean NIP does not cover most adult age groups; hence, the overall vaccination coverage for adults aged 19–64 years is low (<30%), even for those with comorbidities (35–50%) [17,18,19]. The NIP should be expanded to include adults aged 50–64 years and those with comorbidities. Thus, a comparative evaluation for cost-effectiveness is required because adults aged 50–64 years comprise a high percentage of at-risk individuals with chronic diseases, and age-based strategies might be more efficient. 

To address these problems and establish an evidence-based influenza vaccine strategy, we analyzed the cost-effectiveness of various influenza vaccination strategies in three target groups: (1) older adults aged ≥65 years, (2) extended older adults aged 50–64 years, and (3) at-risk adults aged 19–64 years with chronic medical conditions.

## 2. Methods

### 2.1. Vaccination Programs Evaluated

This study simulated four scenarios for older adults aged ≥65 years and three each for adults aged 50–64 years and at-risk adults aged 19–64 years, as described below and as shown in Figure 1. 

#### 2.1.1. Older Adults Aged ≥65 Years

-Program 1 (baseline): all older adults received the trivalent influenza vaccine (TIV) according to the current Korean NIP.-Programs 2, 3, and 4: assume the introduction of a quadrivalent influenza vaccine (QIV), adjuvanted trivalent vaccine (ATIV), or high-dose QIV (HD-QIV) to the NIP instead of the TIV, and target a vaccination rate of 85%.

#### 2.1.2. Adults Aged 50–64 Years and at-Risk Adults Aged 19–64 Years

-Program 1 (baseline): individuals receiving influenza vaccination with out-of-pocket expenses (TIV or QIV).-Programs 2 and 3: assume the introduction of a TIV and QIV, respectively, into the NIP with a target vaccination rate of 80%.

The coverage of the extended program is assumed to reach 80% (adult groups) or 85% (older adults), whereas the baseline program is expected to remain the same as the actual coverage observed in a recent study. In the recent year of the study, the vaccination rate of adults aged 50–64 years was 41.4%, and that of at-risk adults aged 19–64 years was 35.8% [17,19]. Influenza vaccination rates in older adults have been comparatively high (80–85%) since the 2015/2016 season [18]. It was assumed that the TIV and QIV were administered equally in both adult groups, who were not yet counted in the NIP, based on the supply data of the Health Insurance Review and Assessment Service for the corresponding season (unpublished data).

### 2.2. Model Design and Structure

A one-year static decision-tree model was developed for this study. The model structure consists of branching for the clinical and economic outcomes of influenza cases associated with TIV, QIV, or other highly immunogenic influenza vaccine strategies, and subsequent branching of the tree reflects seasonal influenza-related complications, hospitalization, and mortality. Those who were infected but did not visit a clinic were excluded from the model considering their low cost burden and more conservative approach. This model allows the conditional probabilities for each chance node to be defined separately for each cohort (Figure 2).

Given that the model compared only cohorts over one year, discounting was not applied to either cost or outcomes; only productivity loss due to early death from influenza was discounted at 4.5% in accordance [20]. The effects of different discount rates were explored in a sensitivity analysis (0–7.5%). The economic evaluation was conducted from a societal perspective, allowing the inclusion of costs associated with sick leave. A healthcare sector perspective was also considered in the cases of adults aged 50–64 and at-risk adults aged 19–64. The outcome was the incremental cost per quality-adjusted life-year (QALY) gained (incremental cost-effectiveness ratio [ICER]). TreeAge Pro 2020 R2.1 was used to develop the model and perform all analyses.

### 2.3. Input Data

#### 2.3.1. Population Data

Age-specific Korean population estimates and all-cause mortality for 2018 were obtained from the Korean Statistical Information Service (KOSIS) (Table 1) [18]. Several assumptions were required because of the lack of information. The proportion of at-risk adults aged 19–64 years was obtained from a sample cohort of approximately 1 million individuals enrolled in the Korean National Health Insurance Service (NHIS). Based on estimates of the same sample cohort, the model presumed that all-cause mortality in the at-risk group was 3.6-fold higher than that in the general population. The NHIS enrolled almost all citizens in Korea, and the sample cohort is a representative alternative to NHIS data and is potentially used for convenience [21]. The following were defined as at-risk medical conditions: chronic respiratory disease, chronic heart disease, chronic renal disease, chronic liver disease, neurological disease, metabolic syndrome, autoimmune diseases, malignancy, hematologic disease, or immunosuppressed states (ICD-10 codes are presented in Appendix A) [22].

#### 2.3.2. Disease Burden (Probability and Vaccine Data)

A detailed method for estimating disease burden has been previously published [23]. The present analysis differs from the previous one in that it uses the most recent influenza season. Briefly, using the diagnostic code (ICD-10) of the Korean NHIS claims data, the disease burden due to influenza in each target group was estimated for the 2017–18 influenza season. We extracted broad diagnostic codes for hospital visits with influenza-like illness (ILI), rather than the influenza code alone, to obtain the adequate influenza incidence (Appendix A). Based on the ILI surveillance data performed by the Korean Disease Control and Prevention Agency (KDCA), the number of ILI cases was multiplied by 0.281 to estimate the annual incidence of laboratory-confirmed influenza [25]. The probability of hospitalization, complications, and death related to influenza was calculated within the confirmed influenza cases corresponding to the seasonal influenza code (Appendix A). Complications associated with influenza were categorized as “acute complications” and “exacerbation of the existing chronic disease,” as described in a previous analysis [23]. The ICD-10 codes used to define corresponding influenza-related complications are provided in Appendix A. Influenza-related mortality was defined as death within four weeks after the diagnosis of influenza. 

#### 2.3.3. Cost Data

Using the NHIS claims data for the 2017–18 influenza season, per-person medical expenditures were calculated for the following four scenarios: uncomplicated outpatient, complicated outpatient, uncomplicated hospitalization, and complicated hospitalization. Outpatient costs included the cost of office and emergency room visits, including any visits within 14 days of the initial clinic visit, and prescription drug costs at the same time. However, to exclude possible unrelated costs other than those for the treatment of influenza in the outpatient setting, prescription drug costs were excluded for uncomplicated cases; only oseltamivir cost (USD 17.92) was added. Considering that the influenza rapid antigen test is an out-of-pocket expenditure unregistered in the insurance system, influenza rapid antigen test costs (USD 14.98) were added in both outpatient settings. Hospitalization costs comprised costs associated with inpatient hospital stay, outpatient visits, and prescription drugs within 14 days after the influenza diagnosis.

Direct nonmedical costs consisted of transportation and nursing costs. The transportation cost per case of hospitalization or outpatient visit (USD 21.13, 2017) was assigned the value presented in the Korean Health Panel Survey of 2017 [26]. Transportation costs were adjusted to the 2018 prices using the KOSIS transportation cost inflation index (1.0242) [18]. The mode for daily nursing cost (USD 71.68) was the value presented in the raw data of the Korean Health Panel Survey of 2016 (unpublished data), corrected using the KOSIS hospital service inflation index (0.9957) [18]. The nursing cost was multiplied by 0.587 (adult age) or 0.717 (older adults) because only some hospitalized patients received daily nursing care [26]. Nursing costs were applied during the number of visits (outpatient) or length of stay (inpatient). 

The human capital approach was used to estimate indirect costs based on the cost of workdays lost due to illness. Indirect cost was the sum of the cost of workdays lost in visiting clinics or hospitalization, and income lost due to early death. Data from the 2018 survey reports on labor conditions by employment type [27], and the economically active population [18] were used to determine the employment rate and average income (Appendix A). The number of workdays lost was equal to the number of visits (for outpatients) or length of stay (for inpatients). Indirect costs due to early death were calculated using the following formula: $$\mathrm{Cos}t\;\mathrm{of}\;\mathrm{lost}\;\mathrm{income}\;\mathrm{due}\;\mathrm{to}\;\mathrm{early}\;\mathrm{death}\;=\overset{t}{\underset{t=1}{\sum }}\overset{64}{\underset{i=19}{\sum }}\frac{D_{i}P_{i}}{{\left(1+r\right)}^{t}}\;\mathrm{Pi}\;=\mathrm{average}\;\mathrm{yearly}\;\mathrm{income}\;\mathrm{by}\;\mathrm{age}\;$$
t = lost workdays
i = age at time of death
Di = the number of deaths by age
r = discount rate

It was assumed that older adults (aged 65 years or older) incurred no loss of productivity for the patients themselves, as they are not usually employed.

Vaccine costs were assumed to be $8.19 (adult age) or $7.70 (older adults) for TIV, based on the 2018 NIP procurement price at private and public health institutions and the utilization rate of each health institution. Given that the NIP procurement price of the QIV is not yet known, the private market price ratio of the TIV and QIV (1.7 times above) was applied to the NIP procurement price of the TIV to assume that of the QIV. As the ATIV and HD-QIV are not currently available in Korea, an assumed price is also required. Considering the ratio of vaccine price sold in the United States and the price of ATIV in the past in Korea [23,28], it was calculated as twice the TIV price as a base analysis, and 1.5–3 times as a sensitivity analysis. The HD-QIV costs were assumed to be the same as the ATIV, considering the average sales price in the United States [28]. The administration fee was $17.03, which was the same as that of the existing NIP (Table 1). The Korean won was converted into USD at an exchange rate of 1 USD = 1116 KRW. 

#### 2.3.4. Influenza Type or Subtype Circulation Data

A full description of influenza circulation data has been previously published [23]. Based on hospital-based influenza morbidity and mortality (HIMM) surveillance data from 2011 to 2012 to 2015–2016, the average fraction of influenza B was 23.1%. Sensitivity analysis was performed in the range of 9–50%, considering the seasonal variation and KDCA surveillance results [25]. The average TIV influenza B mismatch with the lineage of the vaccine strain was 64.7% (0–97%) [29] (Appendix A). 

#### 2.3.5. Utility Data

The baseline utility data for healthy and at-risk populations were calculated from a Korean national study, adjusting for the different distributions of age groups used in the model [18,24]. As domestic studies on the utility of at-risk groups and disutility due to influenza have not been conducted thus far, they were determined through expert meetings based on foreign results. The baseline utility of the at-risk group was assumed to be 15% lower than that of the general population [23]. The disutility of uncomplicated outpatients, complicated outpatients, uncomplicated hospitalizations, and complicated hospitalizations was −0.35, −0.4, −0.4, and −0.5, respectively. The duration of disutility was calculated as the number of visits (outpatient) or the length of stay (inpatient) (Table 1).

#### 2.3.6. Vaccine Characteristics (Efficacy and Cost of Vaccines)

The vaccine efficacy of TIV against laboratory-confirmed influenza was derived from the Cochrane Database [30,31], with 59% for adults aged 19–64 years and 58% for people aged ≥65 years. Based on previous research and cost-effectiveness studies, we assumed a 20% additional vaccine efficacy for ATIV compared to TIV [13,14,15] and a 24% additional vaccine efficacy for HD-QIV compared to QIV [11,12,16].

The relative advantage of QIV was imputed, similar to that previously reported [32]. Considering the average isolation rate of B strain virus and the level of B lineage mismatch, and assuming a relative efficacy for QIV vs. TIV against a mismatched B lineage of 35% [33,34], it was possible to estimate a surplus in vaccine efficacy (5.2%) for QIV vs. TIV by weighting the probabilities mentioned above. For sensitivity analysis, a wide range of isolation rates of the B strain virus (9%–50%) and mismatching rates (0–97%) were applied. This estimate was applied irrespective of age and medical condition because there is currently no evidence suggesting a difference (Table 2) [35,36]. 

### 2.4. Sensitivity Analysis

To assess the robustness of the results, two sensitivity analyses were performed with major input parameters: a one-way sensitivity analysis to determine which variable had the greatest impact on the cost-effectiveness results and a probabilistic sensitivity analysis (PSA), which assesses the level of uncertainty of the variables in combination within the model. The PSA was performed using a Monte Carlo simulation with 5000 iterations by selecting the input parameter values from a probability distribution (Appendix A). An ICER below the per capita gross domestic product (GDP) is considered cost-effective [37]. The per capita GDP of Korea was $34,767 in 2018.

### 2.5. Consideration of Herd Effect

Herd protection effects were not included in the base case analysis. However, an additional analysis was performed to consider herd effects by calibrating the estimates reported in previous analyses [38,39]. With this approach, the infection rate was projected for groups other than the children’s age by multiplying by the relative risk (RR), which was calculated as follows: $$\mathrm{Formula}\;1:\;RR_{other\;age\;groups}=1-V_{e}$$
$$\mathrm{Formula}\;2:\;RR_{other\;age\;groups}=1-4.6656*V_{e}*P_{children}$$
$$V_{e}=effective\;coverage\;in\;children\;$$
$$P_{children}=\mathrm{proportion}\;\mathrm{of}\;\mathrm{children}$$

If the target vaccination rate (80%) was reached in children compared to the current vaccination rate (55%), the relative risk of infection in other age groups was calculated as 0.92–0.9851 using these formulas [40,41]. Given that the herd effect might vary depending on the age group’s pre-existing immunity, this effect was calculated by applying the current vaccination rate based on the trend shown in a previous study [42].

## 3. Results

For each vaccination strategy, data on the estimated influenza disease burden, cost, and utility are shown in Table 3. 

### 3.1. Older Adults Aged ≥65 Years

#### 3.1.1. Base Case Analysis

In the base case analysis, HD-QIV is expected to be a superior strategy compared to all alternatives, with the lowest cost and highest utility. A comparison with the current vaccination rate (TIV 85%) is shown in Table 4. The QALY parameters in Table 4 indicate that for a threshold of $34,767/QALY, QIV is not expected to be cost-effective (ICER = $46,486/QALY), whereas ATIV is expected to be cost-effective (ICER = $34,314/QALY) compared to TIV. If switched to HD-QIV, the cost reduction due to the reduced influenza disease burden is expected to exceed the cost increase from the high vaccination price, thus resulting in cost savings.

#### 3.1.2. Sensitivity Analysis

Figure 3A and Appendix A show a one-way sensitivity analysis of QIV, ATIV, and HD-QIV compared with TIV among older adults. When switching from TIV to QIV, the ICER was sensitive to vaccine efficacy, price, and incidence. When switching from TIV to ATIV, the ICER was slightly lower than the willingness-to-pay (WTP) threshold; thus, cost-effectiveness was sensitive to all major parameters. For HD-QIV, only one parameter (vaccine cost) had a substantial effect on ICER. Cost-effectiveness varies depending on vaccine cost; switching to QIV is expected to be cost-effective if the cost difference between QIV and TIV is less than $4.79. It is even cost-saving if the cost difference is less than $3.01. In the case of ATIV, switching to ATIV is expected to be cost-effective if the cost difference is from $4.86 to $7.73, and it is even cost-saving if the cost difference is less than $4.86. For HD-QIV, if the cost difference from TIV is within $12.88 or $8.10, switching to HD-QIV is expected to be cost-effective or cost-saving, respectively. 

In the PSA, where the vaccine cost was fixed and all other major parameters were changed, 67.9%, 79.2%, and 92.2% of the simulations fell below the one-GDP per capita WTP threshold for the QIV, ATIV, and HD-QIV strategies, respectively (Figure 4A and Appendix A). Considering the indirect herd effect, switching to QIV is not expected to be cost-effective. However, if the cost difference between the two vaccines is less than $4.40, this strategy could be cost-effective even if the herd effect is applied to the maximum. Switching to ATIV was not cost-effective under the maximum herd effect. However, even in such cases, ATIV is expected to be cost-effective if the cost difference between the two vaccines is within $7.13. Switching to HD-QIV is expected to be cost-effective regardless of the herd effect.

### 3.2. Adults Aged 50–64 Years

#### 3.2.1. Base Case Analysis

A comparison with the current vaccination strategy is shown in Table 4. From a societal perspective, introducing the influenza vaccine NIP targeting a coverage rate of up to 80% is expected to be cost-saving with TIV and cost-effective ($3661/QALY) when changing to QIV. 

However, from the healthcare sector perspective, the introduction of TIV or QIV into the NIP (targeting a coverage rate of up to 80%) is not expected to be cost-effective (ICER $37,352/QALY or $86,463/QALY, respectively) (Table 4).

#### 3.2.2. Sensitivity Analysis

Considering the current trend in South Korea, it is highly likely that QIV will be adopted if NIP is introduced to the expanded adult age group. Thus, a one-way sensitivity analysis was conducted for strategy switching from baseline to QIV (80%), from a societal perspective (Figure 3B). Influenza incidence and vaccine cost had the greatest influence on changes in the ICER values. Switching to QIV is expected to be cost-effective if the cost of QIV is less than $33.48, and even cost-saving if the cost is less than $28.22. In the PSA, in which the vaccine cost was fixed and all other parameters were changed, 67.5% of the simulations were under the WTP threshold (Figure 4B). 

Considering the herd effect, implementing TIV or QIV into the NIP is still expected to be cost-saving or cost-effective compared to the current state, even under the maximum herd effect (Appendix A).

### 3.3. At-risk Adults Aged 19–64 Years

#### 3.3.1. Base Case Analysis

A comparison with the current vaccination strategy is shown in Table 4. From a societal perspective, introducing the influenza vaccine NIP targeting a coverage rate of up to 80% is expected to result in cost savings, regardless of vaccine formulation (TIV or QIV). However, the introduction of the QIV is thought to be a better alternative because of its greater utility gains.

Even from the healthcare sector perspective, the introduction of TIV into the NIP (targeting a coverage rate of up to 80%) is expected to be cost-effective (ICER $23,020/QALY). In comparison, the introduction of a QIV into the NIP is not anticipated to be cost-effective (ICER $53,050/QALY) from this perspective (Table 4).

#### 3.3.2. Sensitivity Analysis

The results of the one-way sensitivity analysis (in terms of ICERs) from a societal perspective are presented in Figure 3C. The cost-effectiveness of the base case remained robust irrespective of changes in any of the major parameters. In the PSA, in which the vaccine cost was fixed and all other variables were changed, 100% of the simulations were below the WTP threshold (Figure 4C). 

Even when the herd effect is applied to the maximum, introducing either TIV or QIV into the NIP (targeting a coverage rate of up to 80%) is expected to be cost-saving compared to the current state.

## 4. Discussion

This study estimated the disease burden of influenza using the NHIS claims data to assess the clinical and economic impact of various influenza vaccination strategies for three target groups: older adults (≥65 years), adults (50–64 years), and at-risk adults (19–64 years). The results showed that HD-QIV had the most favorable cost-effectiveness profile among all existing commercially available alternatives for older adults, based on currently available data. A universal substitution of TIV with ATIV in the Korean older adult population is also expected to be cost-effective given an ICER of less than $34,767/QALY. Regarding other adult age groups (aged 50–64 years and at-risk adults aged 19–64 years), introducing the influenza vaccine into the NIP is expected to be cost-effective, notwithstanding the vaccine formulation, from a societal perspective. 

The results of this study are consistent with previous cost-effectiveness analyses of influenza vaccines targeting adults and older adults. Research has shown that influenza immunization programs, especially those adopting QIV, would be cost-beneficial for those with underlying illnesses, even though one of them assumed an influenza vaccine efficacy as low as 32% [35,43,44]. Moreover, most studies conducted in older adults have suggested that highly immunogenic vaccine formulations are cost-effective compared to standard-dose non-adjuvanted vaccines, although data on the effectiveness of these formulations are limited [11,12,13,32,45]. Based on these results, in most countries, influenza vaccination is supported at the national level for chronically ill individuals [3,4,5,6]. In the case of older adults, highly immunogenic vaccines are also supported in some countries, including the US and the UK [3,4].

However, unlike the research conducted so far [23,44,46], this study showed that replacing TIV with QIV was not cost-effective in older adults in the base case analysis, which assumed the cost difference between the two vaccines to be $5.39. However, the results of the base case were sensitive to the vaccine cost. Since the 2020/21 season in South Korea, a universal substitution of TIV with QIV has been adopted in the NIP, and its price was determined to be within $1.2 of TIV. Therefore, the current changes in the Korean NIP for older adults are considered cost-effective or even cost-saving. Although the influenza vaccine in the NIP has been switched from TIV to QIV, the cost-effectiveness of TIV and QIV remains controversial and can vary greatly depending on the price and disease burden of the target group. Therefore, this study is expected to provide useful information to determine the acceptable price range of vaccines in the NIP and expand the influenza program to other target groups that have not yet been introduced in the NIP. 

Thus far, data on the effectiveness of adjuvanted and high-dose vaccines are limited, and the vaccine price is also dependent on estimates. In this study, the efficacy of the high-dose vaccine was determined to be slightly higher than that of the adjuvanted vaccine based on the available literature. The prices of the two vaccine formulations were the same based on overseas sales prices. Therefore, the high-dose vaccine showed a more favorable cost-effective profile than the adjuvanted vaccine. The cost-effectiveness of adjuvanted vaccines should be re-evaluated after comparison with other vaccines, and the Korean price is established. Nevertheless, estimating the acceptable price range of highly immunogenic vaccine formulations compared to the existing ones (TIV or QIV) provides useful information for future vaccine introduction. This study suggests that switching to HD-QIV is cost-effective when the price difference from TIV is less than $12.89 at the currently applied level of vaccine efficacy (relative risk 0.76 compared to TIV). Switching to ATIV is also cost-effective when the price difference from TIV is less than $7.73 at the currently applied level of vaccine efficacy (relative risk 0.8 compared to TIV).

For the expanded adult age group (50–64 years old) and the at-risk adult group, the introduction of the influenza vaccine NIP targeting a vaccination rate of up to 80% is expected to be cost-saving compared to the current state. However, from the healthcare sector perspective, only the introduction of TIV for at-risk 19–64-year-old adults was found to be cost-effective. When expanding NIP within a limited healthcare budget, costs and benefits within the healthcare system might be more important than indirect costs (productivity loss). In addition, several factors must be considered when deciding the priority between the two adult groups
1.An “age-based” strategy might be a more efficient option considering the wide range of comorbidities, uncertainty of diagnosis, and implementation issues with influenza vaccine administration within a short period (1–2 months). 2.The cost of the government’s investment in the vaccine program ranged from $242 million (TIV 80%) to $297 million (QIV 80%) for 50–64-year-old adults and from $114 million (TIV 80%) to $140 million (QIV 80%) for at-risk 19–64-year-old adults. Therefore, the total cost to be invested is expected to be lower for the at-risk adult group than for the 50-to-64-year-old group, but implementation issues should be considered.3.From a societal perspective, introducing the influenza vaccine into the NIP (TIV or QIV) appears to be a cost-effective or even cost-saving strategy for both adult age groups. However, after excluding productivity loss, the ICER of introducing QIV into the NIP was $86,463/QALY for 50–64-year-old adults and $53,050/QALY for at-risk 19–64-year-old adults, which is no longer cost-effective for both groups with respect to GDP per capita. Considering that the ICER of at-risk 19–64-year-old adults was lower than that of the comparison group, it might be appropriate to expand the immunization program to the at-risk population first. Of course, it would be difficult to directly determine the dominance between the target groups with an ICER only derived from different analyses; however, it might be meaningful because the analytic results were derived from the same data source and analytic methods. 


As expected and evident from the one-way sensitivity analyses, certain key parameters influenced the analysis results. In most analyses, vaccine cost magnified the incremental costs per QALY. As cost is a controllable factor, the findings of this study, which estimate the diverging point of vaccine cost that affects cost-effectiveness, are expected to be useful in policymaking. Vaccine efficacy also appeared to play an important role in the results of each model. Considering that the vaccine efficacy of QIV varied according to the proportion of influenza B and vaccine mismatch, the disease burden of influenza B could also be regarded as a major parameter. In addition, the range of influenza incidence also affects the results; cost-effectiveness results may vary depending on the disease burden of the corresponding season. 

This study had several limitations. First, there are primary limitations concerning the accuracy of estimating the incidence of influenza from claims data using ICD-10 codes. However, these results are consistent with those obtained in previous studies [23,47]. The additional uncertainty was addressed by performing sensitivity analyses within adequate ranges. Second, we could not sufficiently reflect seasonal variability in the NHIS claims data because we used data from only one influenza season. However, across several influenza seasons from 2010–2018, the influenza epidemic size was not a quiet variable. In the 2010–2013 (three seasons) and 2013–2015 (two seasons) data, the influenza incidence in the older adults was 5.52% and 4.9–6.9%, respectively, which is similar to the 5.31% observed in our study [23,48,49]. In this study, seasonal variations were considered in the sensitivity analyses; the variation in the burden of influenza B was allowed in the model using average data over eight seasons in the base case. Third, although some countries switched from ATIV to adjuvanted QIV (AQIV), most nations have not yet introduced AQIV. Moreover, the ATIV was expected to be introduced in South Korea at the time this study commenced. The results of this study are useful because the ATIV strategy in older adults is more cost-effective than the QIV strategy, which means that switching from QIV to the adjuvanted vaccine (ATIV or AQIV) would be reasonable. Fourth, given that the data on highly immunogenic vaccine formulations are largely uncertain, careful interpretation is essential. Nevertheless, the results of this study provide important data for cost selection and cost-effectiveness evaluation when introducing new formulations in the future. Finally, this study did not analyze older adults from the healthcare sector perspective. However, because the productivity loss in older adults is considered minimal, there might be no difference in the cost-effectiveness results from the healthcare sector and societal perspectives. 

In conclusion, the introduction of the influenza vaccine NIP (TIV or QIV) is expected to be cost-effective in the expanded adult age group (50–64-years old) and the at-risk group (19–64-years old). Moreover, this study indicates that highly immunogenic vaccines for older adults are likely to be favored over the standard non-adjuvanted vaccine, based on currently available data. The relative cost-effectiveness of such formulations (ATIV and HD-QIV) should be re-evaluated after establishing their effectiveness and the Korean price.

## Figures and Tables

**Figure 1 vaccines-10-00445-f001:**
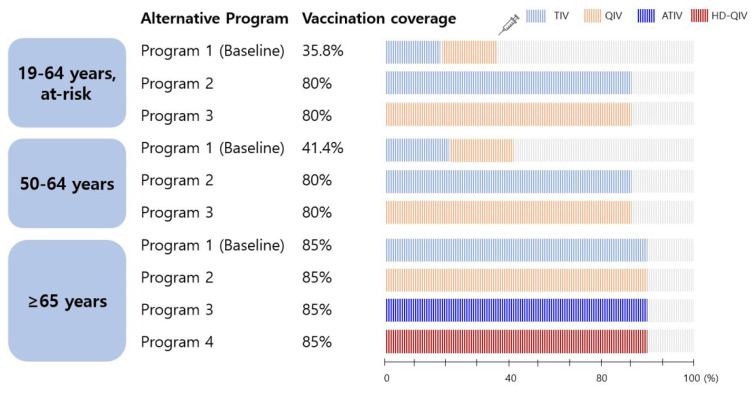
Scenarios simulated in the model. TIV, trivalent influenza vaccine; QIV, quadrivalent influenza vaccine; ATIV, adjuvanted trivalent influenza vaccine; HD-QIV, high-dose quadrivalent influenza vaccine.

**Figure 2 vaccines-10-00445-f002:**
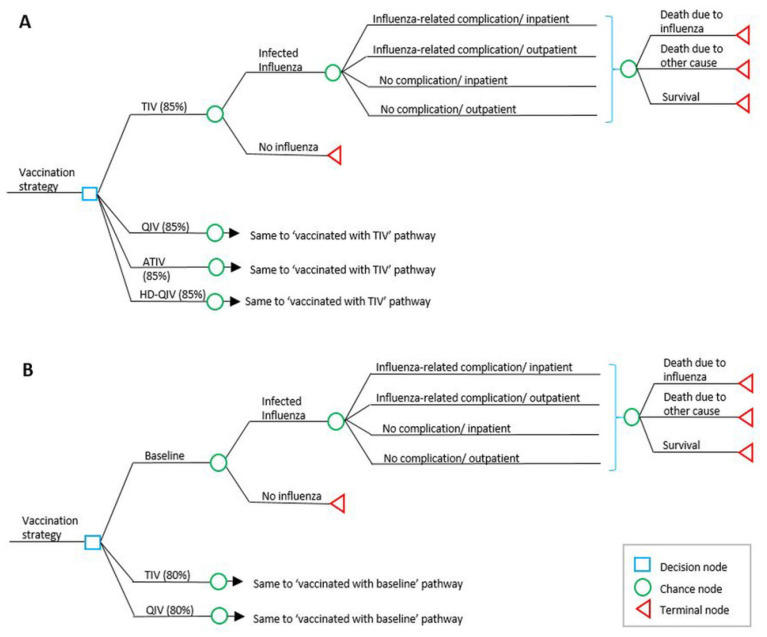
(**A**): Overview of decision tree model structure in older adults; (**B**): model structure in individuals aged 50–64 years and at-risk adults aged 19–64 years. TIV, trivalent influenza vaccine; QIV, quadrivalent influenza vaccine; ATIV, adjuvanted trivalent influenza vaccine; HD-QIV, high dose quadrivalent influenza vaccine.

**Figure 3 vaccines-10-00445-f003:**
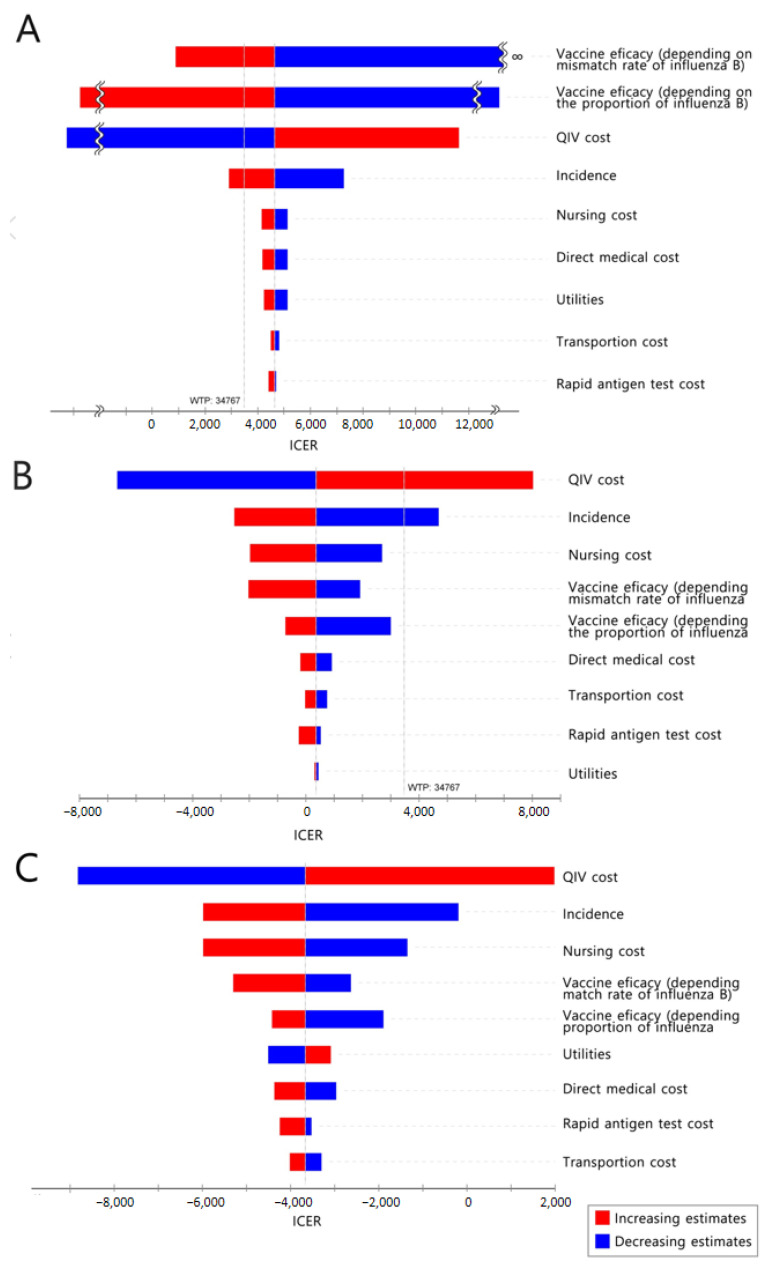
One-way sensitivity analysis. (**A**): QIV compared to TIV in older adults; (**B**): QIV compared to the current state in adults aged 50–64 years; (**C**)**:** QIV compared to the current state in at-risk adults aged 19–64 years. TIV, trivalent influenza vaccine; QIV, quadrivalent influenza vaccine; WTP, willingness to pay; ICER, incremental cost-effectiveness ratio.

**Figure 4 vaccines-10-00445-f004:**
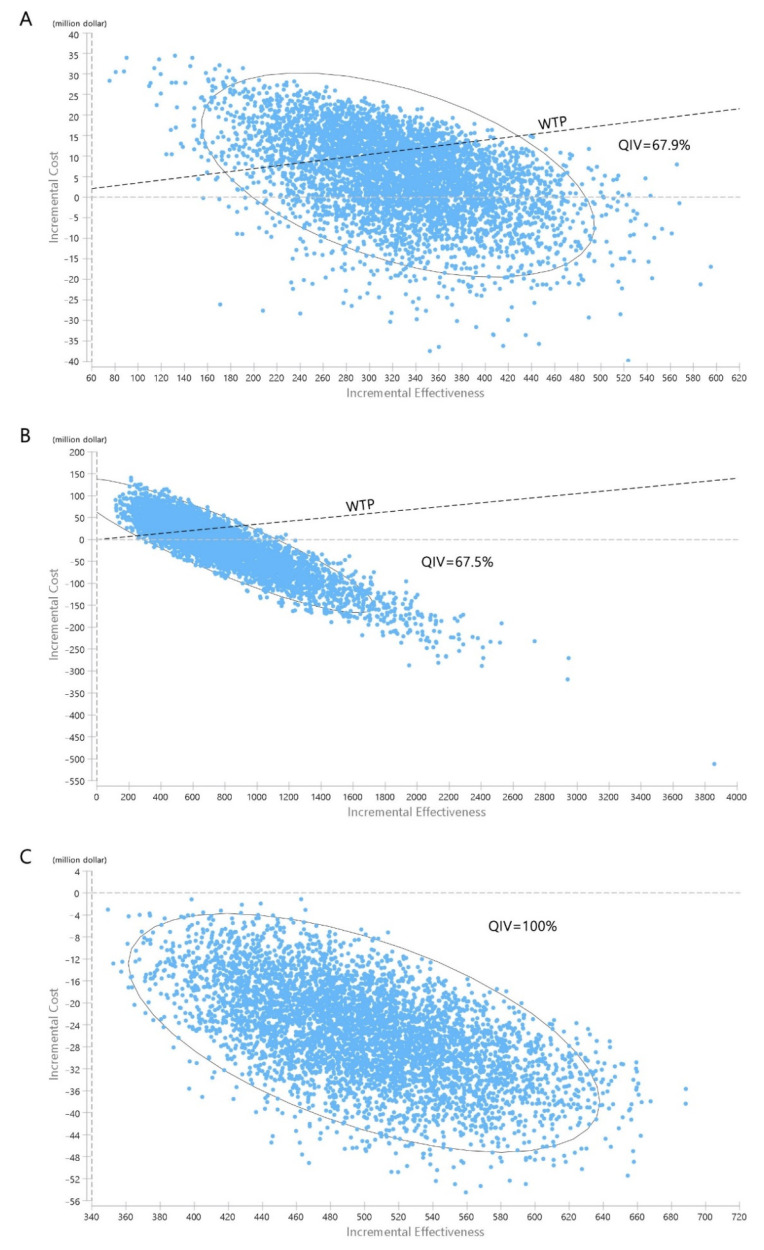
Probabilistic sensitivity analysis. (**A**): QIV compared to TIV in older adults; (**B**): QIV compared to the current state in adults aged 50–64 years; (**C**): QIV compared to the current state in at-risk adults aged 19–64 years. TIV, trivalent influenza vaccine; QIV, quadrivalent influenza vaccine.

**Table 1 vaccines-10-00445-t001:** Input data for disease probability, cost, and utility.

Parameter	Target Groups			Reference
	19–64 Years, At-Risk	50–64 Years	≥65 Years	
Population	5,636,765	11,998,063	7,455,149	[18], assumption
Probability of disease burden				Extracted from NHIS database
Influenza case	0.0549	0.0454	0.0531	
Complication/Influenza case	0.0675	0.0515	0.1417	
Hospitalization/Influenza case	0.0638	0.0553	0.1046	
Death/Influenza case	0.000349%	0.000232	0.005740	
Costs (USD) *				
Vaccination				
TIV	25.22	25.22	22.86	Government data
QIV	30.95	30.95	28.24	Assumption
ATIV	-	-	30.55 (26.70–38.25)	Assumption
HD-QIV	-	-		
Out-of-pocket ^a^	31.36	31.36	-	Assumption
Direct medical cost				
Uncomplicated outpatient	61.01	59.90	60.69	Extracted from NHIS database
Complicated outpatient	129.04	118.78	119.60	
Uncomplicated hospitalization	934.17	870.72	1361.41	
Complicated hospitalization	1948.93	1700.13	2278.45	
Direct non-medical cost				
Nursing cost	41.90	41.90	51.18	
Transportation cost	21.64 per episode			
Length of stay (or number of visits)				
Uncomplicated outpatient	3.59	3.66	3.69	Extracted from NHIS database
Complicated outpatient	4.96	5.12	4.36	
Uncomplicated hospitalization	7.66	7.74	8.60	
Complicated hospitalization	11.34	11.00	12.42	
Utility				[18,23,24]
Baseline utility	0.819	0.938	0.867	
Uncomplicated outpatient	−0.35	−0.35	−0.35	
Complicated outpatient	−0.4	−0.4	−0.4	
Uncomplicated hospitalization	−0.4	−0.4	−0.4	
Complicated hospitalization	−0.5	−0.5	−0.5	

* 1 USD = 1116 KRW; TIV, trivalent influenza vaccine; QIV, quadrivalent influenza vaccine; ATIV, adjuvanted trivalent influenza vaccine; HD-QIV, high-dose quadrivalent influenza vaccine; NHIS, National Health Insurance Service. ^a^ For the non-NIP target group, the vaccination cost was calculated based on the market price. Assuming that the TIV and QIV were administered equally, the average prices of TIV and QIV sold in the market were applied.

**Table 2 vaccines-10-00445-t002:** Vaccine efficacy and coverage rates.

		Target Groups			
		At-Risk		50–64 Years	≥65 Years
		19–49 Years	50–64 Years		
Vaccine efficacy	TIV	59%	59%	59%	58%
	QIV	64.2% (59–70.3%)	64.2% (59–70.3%)	64.2% (59–70.3%)	63.2% (58–69.3%)
	ATIV				66.4% (62.2–74.8%)
	HD-QIV				72.0% (68.1–76.7%)
Vaccination coverage rates		35.8%	35.8%	41.4%	84.3%

TIV, trivalent influenza vaccine; QIV, quadrivalent influenza vaccine; ATIV, adjuvanted trivalent influenza vaccine; HD-QIV, high-dose quadrivalent influenza vaccine.

**Table 3 vaccines-10-00445-t003:** Expected clinical outcomes for each vaccine strategy.

≥65 Years					
		TIV (85%)	QIV (85%)	ATIV (85%)	HD-QIV (85%)
Number of	Vaccinated	6,336,877	6,336,877	6,336,877	6,336,877
	Influenza cases	392,724	358,486	300,546	337,417
	Complications	55,649	50,797	42,587	47,812
	Hospitalizations	41,079	37,498	31,437	35,294
	Deaths	2254	2058	1725	1937
Total life-year (QALY)		6,457,913	6,458,238	6,458,437	6,458,786
Total cost (USD)		363,530,403	378,601,928	381,051,836	360,969,390
**50–64 years**					
		**Current**	**TIV (80%)**	**QIV (80%)**	
Number of	Vaccinated	4,967,198	9,598,450	9,598,450	
	Influenza cases	544,712	380,565	350,581	
	Complications	28,053	19,599	18,055	
	Hospitalizations	30,123	21,045	19,387	
	Deaths	126	88	81	
Total life-year (QALY)		11,251,489	11,252,146	11,252,271	
Total cost (USD), societal perspective		554,543,223	524,733,031	557,407,544	
Total cost (USD), healthcare sector perspective		367,995,027	392,517,433	435,608,932	
**19–64 years, at-risk**					
		**Current**	**TIV (80%)**	**QIV (80%)**	
Number of	Vaccinated	2,017,962	4,509,412	4,509,412	
	Influenza cases	309,564	207,218	190,892	
	Complications	20,896	13,987	12,885	
	Hospitalizations	19,750	13,221	12,179	
	Deaths	108	72	67	
Total life-year (QALY)		3,982,199	3,982,569	3,982,630	
Total cost (USD), societal perspective		306,667,009	278,606,389	288,380,562	
Total cost (USD), healthcare sector perspective		189,104,801	198,971,911	215,629,718	

**Table 4 vaccines-10-00445-t004:** Base case analysis (per-person cost and effectiveness).

		Cost (USD)	Incremental Cost (ΔUSD)	Effectiveness (QALY)	Incremental Effectiveness (ΔQALY)	ICER (ΔUSD/ QALY)
≥65 years	TIV	363,530,403		6,457,913		(reference)
	QIV	378,601,928	15,071,525	6,458,238	324	46,486
	ATIV	381,501,836	17,971,433	6,458,437	524	34,314
	HD-QIV	360,969,390	−2,561,013	6,458,786	873	Cost-saving
50–64 years	Current	554,543,223		11,251,489		(reference)
	TIV	524,733,031	−29,810,192	11,252,146	657	Cost-saving
	QIV	557,407,544	2,864,321	11,252,271	782	3661
19–64 years, at–risk	Current	306,667,009		4,614,862		(reference)
	TIV	278,606,389	−28,060,620	4,615,290	429	Cost-saving
	QIV	288,380,562	−18,286,447	4,615,361	500	Cost-saving
**From healthcare sector perspective**						
50–64 years	Current	367,995,027		11,251,489		(reference)
	TIV	392,517,433	24,522,405	11,252,146	657	37,352
	QIV	435,608,932	67,613,905	11,252,271	782	86,463
19–64 years, at–risk	Current	189,104,801		4,614,862		(reference)
	TIV	198,971,911	9,867,110	4,615,290	429	23,020
	QIV	215,629,718	26,524,917	4,615,361	500	53,050

## Data Availability

The data presented in this study are available within the article or in the Appendix A.

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
