# Peer review of "Cost-Effectiveness of Influenza Vaccination Strategies in Adults: Older Adults Aged ≥65 Years, Adults Aged 50–64 Years, and At-Risk Adults Aged 19–64 Years"

_vaccines, 2022, doi:10.3390/vaccines10030445_

Round 1

Reviewer 1 Report

Dear Editor and authors,

the topic of the manuscript (enlargement of influenza vaccination recommendations) is certainly interesting and greatly discussed in many countries. The paper is well written. The mathematical model is well performed but some input data and assumptions should be reviewed and update. This analysis appears performed some years ago and must be update considering the current available vaccines and national recommendations. Globally, the manuscript should be reconsidered after major revision.

You find now some suggestions to authors:

  • Line 44: in some countries influenza vaccination is only greatly recommended but not mandatory. Please include this possibility together with mandatory indication.
  • Figure 1: Please specify in the figure types of vaccine assumed in the different programs in order to facilitate the reading.
  • Line 86: the study considered the switch from TIV to ATIV. Now AQIV is available in many countries. Why the switch to AQIV was not analyzed the manuscript? Not include this vaccine makes this study not up to date enough.
  • Line 89-90, program 1 Adults Aged 50–64 Years and At-risk Adults Aged 19–64 Years: what types of vaccines are planned for this scenario? TIV and QIV? With what proportion? It is not clear.
  • Line 107-108, “Those who were infected but did not visit a clinic were excluded from the model”: please specify the rationale of this assumption. The analysis is performed for the societal perspective and therefore, also these subjects could have some (small) costs due to influenza.
  • Line 117-121: please specify because the analysis for elderly was performed only from societal perspective while in the other two groups also from healthcare perspective. Results for healthcare perspective in the elderly should be showed, especially if this perspective is considered in the decision process.
  • Line 148-163: the incidence of ILI is greatly variable in the different seasons due to genetic variability of virus, past circulation of specific virus, vaccine coverage, mismatch, ….. Therefore, it is preferable use average epidemiological data of previous influenza seasons in order to reduce this variability that could influence the results of the analysis and not only perform a sensitive analysis. In addition, what were the characteristics (for example, intensity) of the 2017-2018 season compared to the other seasons? Also, the rate of laboratory-confirmed influenza should be derived from the average of data of some seasons.
  • Line 350-359: specify that these results referred to societal perspective.
  • Line 374-375: based on what the authors wrote, since the 2020/21 season in South Korea a universal substitution of TIV with QIV has been adopted in the NIP. Therefore, the comparator of the economic analysis should be QIV and not TIV (that should be excluded from the analysis), otherwise the analysis is old, not updated and not useful.
  • Line 378-392: as previously suggested, AQIV should be included in the analysis.
  • Line 393-395: this sentence is true only from the societal perspective. Please specify it in the text. The not favorable economic results from healthcare perspective can influence the health decisions on vaccination in Korea? Is only data from the societal perspective considered in the decisional process?

Author Response

Please see the attachment (Please note that the changes in page numbers in this document are as per the revised manuscript [simple version])

Reviewer 2 Report

The manuscript titled "Cost-effectiveness of influenza vaccination strategies in adults: older adults aged ≥65 years, adults aged 50–64 years, and at-risk adults aged 19–64 years" deals with a very interesting topic. The cost-effectiveness evaluation is an important step for the introduction of a new vaccine or a new target.

The paper is well-written and the methodology is explained in detail.

However, the references should be updated with the most recent papers. Above all, those relating with the efficacy of highly immunogenic influenza vaccines, such as high-dose or with MF59-adjuvanted vaccines, must be updated and I suggest to modify the sentences in Lines 57-58 and 435-436 with the data on the efficacy of these vaccines.

In addition, the authors may consider the following minor comments:

Line 306: the ICER of QIV in  group "50-64 years" is reported as $86,486 but, in Table 4, this value is 86,463. Please, resolve this inconsistency.

Line 408: the ICER of QIV in group "at risk 19-64 years" is reported as $53,053 but, in Table 4, this value is 53,050. Please, resolve this inconsistency.

Author Response

(The authors gave the same response as above.)
